# An Unusual Case of Erdheim Chester Disease (ECD) with Knee Pain: A Case Report

**DOI:** 10.3390/medicina59071288

**Published:** 2023-07-12

**Authors:** Yong Bum Joo, Young Mo Kim, Woo Yong Lee, Kun Woo Lee, Hyung-Jin Chung

**Affiliations:** 1Department of Orthopedic Surgery, Chungnam National University Hospital, Chungnam National University College of Medicine, Daejeon 35015, Republic of Korea; 2Department of Orthopedic Surgery, Chungnam National University Sejong Hospital, Chungnam National University College of Medicine, Sejong 30099, Republic of Korea

**Keywords:** Erdheim Chester disease, histiocytosis, knee pain, case report

## Abstract

*Background:* Erdheim Chester disease (ECD) is a rare, non-Langerhans cell histiocytosis of unknown etiology that occurs in multiple organs. The clinical characteristics of ECD are unknown, making it difficult to diagnose. *Case presentation:* A 61-year-old woman presented with left knee pain and contracture. She had recent medical problems such as recurrent urinary tract infection, pericardial effusion, and pleural effusion. Simple radiography and magnetic resonance imaging of the knee revealed an osteosclerotic lesion. Under suspicion of malignancy, other radiologic modalities were performed, but there were no significant results showing malignancy. A bone biopsy of the knee lesion led to a final diagnosis of ECD. The patient was treated with systemic steroids and was ultimately tried on PEG-interferon. *Conclusion:* This report describes an unusual presentation of ECD involving the skeletal system and multiple extraskeletal organs. Owing to its non-specific nature, ECD was notably difficult to diagnose. Therefore, if a patient has knee pain and other multiorgan presentations without malignancy, clinicians should suspect ECD.

## 1. Background

Erdheim Chester disease (ECD) is a very rare non-Langerhans cell histiocytosis (LCH) that occurs in the skeletal system and multiple extraskeletal organs [1,2,3,4]. Fewer than 1000 cases of ECD have been reported worldwide [1]. In Korea, the first case of ECD was reported by Park et al. in 1999, and 34 cases have since been reported [5]. 

ECD infiltrates the bones and various organs via lipid-laden histiocytes that have a foamy or eosinophilic cytoplasm [2,3,4]. Lipid-laden histiocytes produce xanthogranulomas, which can be found in many internal organs and tissue sites, including the long tubular bones, skin, lung, pericardium, kidney, retroperitoneal space, orbit, and brain [2,4].

ECD is a progressive disease involving multiorgan dysfunction, associated with poor prognosis [2,3,6,7]. Patients with ECD exhibit several symptoms that depend on the organ involved. Therefore, ECD is difficult to diagnose, often undiagnosed, or misdiagnosed as LCH or multiple bone metastasis [3]. ECD mostly affects adults; the average age of the diseased population is 53 years, with a slight male predominance for unknown reasons [1,4,8]. Most patients have skeletal involvement at the time of diagnosis, and extraskeletal involvement occurs in approximately 50% of patients [2,3].

There is no standardized treatment for ECD, but treatment has progressed dramatically because of a better understanding of the molecular aspects of ECD. Various treatment regimens, including some conventional medicines and, more recently, targeted therapies, have been tried [9].

In this article, we report a typical but difficult case of ECD that was undiagnosed until the authors performed a bone biopsy. In this report, in comparison with previous cases to determine differences, we discuss the diagnostic tools and approach to the disease. This paper aims to raise the awareness of ECD.

## 2. Case Presentation

A 61-year-old woman was referred to the orthopedic department for mild knee pain and left knee contracture that began four months previously. She had no trauma history and complained of knee pain gradually. On physical examination of the knee, a mild effusion was detected and she complained of broad pain around the knee. The range of motion was 20–160 degrees, and mechanical symptoms were not prominent. She had a history of hypertension, chronic kidney disease, and diabetes mellitus. Within two years previously, she had been admitted to the internal medicine department of three different hospitals for recurrent urinary tract infection, pleural effusion, and pericardial effusion.

For knee pain and contracture, radiologic evaluation (simple radiography and magnetic resonance imaging (MRI)) of the knee joint was performed at another hospital, and the findings showed osteosclerotic lesions, which were thought to be bone metastases; therefore, the patient was referred to the hemato-oncology department of our hospital for cancer evaluation (Figure 1).

Before a bone biopsy, serum and pleural fluid analyses for hemato-oncologic disease, chest computed tomography (CT), abdominal CT, pericardial biopsy by a cardiothoracic surgeon, and whole-body positron emission tomography-CT (PET-CT) were performed for cancer evaluation. Based on a suspicion of malignancy, serum evaluation for cancer antigen (CA)-15-3 and pleural fluid evaluation for carcinoembryonic antigen (CEA) were performed, and the results were within the normal range (CA15-3, 7.54 U/mL; CEA, 0.88 ng/mL).

The chest CT showed notable pericardial effusion with pericardial enhancement and pleural effusion with pleural enhancement (Figure 2). The abdominal CT showed fluid collection in both perirenal spaces and diffuse infiltration in both kidneys (Figure 3). Pericardial biopsy revealed chronic active inflammation with nonspecific findings. The PET/CT showed increased uptake in the sacrum, distal femur, proximal tibia, and spleen, with pericardial and pleural effusion (Figure 4). The simple radiography and MRI of the left knee showed patchy sclerotic changes in the medulla of the distal femur and proximal tibia (Figure 1).

Based on these findings, hematologic malignancy, such as lymphoma or bone metastasis, was suspected, but no diagnostic tools confirmed the patient’s disease. Therefore, we performed a bone biopsy of the anterior cortex and medulla of the proximal tibia. The bone biopsy revealed the identity of the disease by showing diffuse infiltration of foamy histiocytes in the marrow space. Immunohistochemical staining was positive for CD68 and negative for S-100 and CD1a (Figure 5).

After the bone biopsy, a whole-body bone scan (WBBS) and genomic DNA analysis were performed for further evaluation. The WBBS results showed diffuse uptake in the tibia, femur, radius, and ulna, which is a typical finding of bone marrow involvement in ECD (Figure 6). Genomic analysis showed a *BRAF* mutation, which is also a typical finding of ECD (Figure 5). We diagnosed the patient with ECD based on the clinical information, radiologic findings, and histologic findings. The patient is currently undergoing steroid treatment in the hemato-oncology department and is preparing to apply PEG-interferon. 

## 3. Discussion 

The clinical characteristics of ECD are unknown, nonspecific, and difficult to diagnose. Owing to the progressive nature of ECD, early diagnosis and treatment are critical to patient prognosis. Therefore, a clinical suspicion of the disease is very important. This section aims to describe the features that clinicians should be aware of to avoid misdiagnosing or not diagnosing ECD in order to obtain promising results.

The clinical manifestations of ECD include arthralgia, keloid formation, and systemic symptoms due to the involvement of several organs such as the heart, liver, lungs, and kidneys [3,10,11]. Skeletal involvement is present in most cases, with osteosclerotic lesions in bilateral symmetrical patterns in the metaphysis of long bones. Skeletal involvement of the ribs, sacrum, head, neck, and spine can also occur [3,12]. Extraskeletal involvement occurs in approximately 50% of patients [2,3]. Patient prognosis is determined by the affected organs; if the blood vessel system is affected, the patient has a poor prognosis [3,5,6].

Cavalli et al. [8] recently reviewed all published cases and identified 259 ECD patients. According to their report, as in our case, knee involvement occurred in approximately 16% of patients with skeletal involvement, and they explained symptoms such as pain, swelling, and a palpable mass.

ECD can lead to fatal results without appropriate treatment. As discussed above, ECD can be clinically misdiagnosed or undiagnosed because patients with arthralgia, contracture, or lower extremity pain are often observed with orthopedic osteoarthritis [3,12]. Therefore, if the disease is not recognized, it is impossible to prevent the progression of ECD, and the prognosis is poor.

Our patient showed typical features of patchy-sclerotic lesions in the bilateral symmetrical boundary with involvement of several organs, such as the kidneys, heart, and lungs. Clinically, these bony lesions can be misdiagnosed or confused with multiple bone metastases without suspicion of ECD. In our patient’s case, hemato-oncologists were the first clinicians to see this patient, and they suspected the patient had bone metastasis before the orthopedist performed a bone biopsy. The histological findings of the bone biopsy revealed her disease as ECD. The patient was positive for CD68 upon immunohistochemical staining, which is characteristic of LCH, but negative for CD1a and S-100. From these results, LCH can be excluded from the diagnosis [3].

ECD can be diagnosed using a combination of orthopedic bone biopsies, radiologic findings, and clinical characteristics. A bone biopsy is the most important diagnostic tool to obtain a definitive diagnosis; therefore, orthopedic surgeons play a critical role in the diagnosis of ECD. After the diagnosis, patients can be treated systemically to prevent disease progression. However, recognizing ECD is difficult without suspicion, which makes ECD a fatal disease with a poor prognosis. Therefore, being aware that ECD can occur is crucial.

Because of the rarity of ECD and a lack of previous studies, optimal treatment has yet to be established [2]. Patients with this disease have undergone treatment with various cytotoxic agents, systemic steroids, radiation therapy, and autologous stem cell transplantation [13]. Systemic steroids rapidly reduce inflammatory reactions; they are reserved for patients who cannot tolerate more aggressive therapies but are not effective long-term as monotherapy. Radiation therapy may be suitable for local palliation. Some studies have reported that interferon-α or Pegylated interferon-α is a valuable treatment for ECD patients. The clinical response to interferon-α or Pegylated interferon-α is variable, and the dose and duration of therapy are uncertain, but in general, treatment improves disease burden and survival [14]. If interferon-α fails, imatinib mesylate has been suggested as a potential experimental alternative therapy by some reports [15]. Furthermore, treatment with a *BRAF* inhibitor has been considered to be a helpful therapy, as individuals with multisystemic and refractory ECD and *BRAF* V600E gene mutations have shown partial clinical responses following treatment with a *BRAF* inhibitor [9,16]. 

In our case, we described a typical case of ECD. However, there are two aspects that distinguish our case from other previous cases. First, our case showed not only pain, but also proceeded with contracture. The contracture was not simply due to pain; it showed a worsening pattern due to mechanical problems such as deformities of the ligament, joint capsule, and bone. Moreover, in the case of our patient, the diagnosis could have been made earlier, but was not. For the knee pain and contracture, radiologic evaluation of the knee joint was performed at another hospital and the findings showed osteosclerotic lesions, which were thought to be bone metastases; therefore, the patient was referred to the hemato-oncology department of our hospital for cancer evaluation. In the process of cancer evaluation, there was a delay in consultation with the orthopedic surgery department with regard to the knee pain. Even so, we thought it pertinent to suspect ECD at the time of consultation and actively performed bone biopsies, which may be another distinguishable factor of our case.

To summarize, if a middle-aged patient has arthralgia, knee contracture, or lower extremity pain and clinical symptoms involving multiple organs for unknown reasons, radiologic evaluation of skeletal symptoms should be performed. If the radiology shows symmetric and bilateral bone lesions and the patient has no sign of malignancy, a clinical suspicion of ECD is necessary, and an orthopedic bone biopsy should be performed. After diagnosis, an orthopedic surgeon can refer the patient to a hemato-oncologist for appropriate management. 

## 4. Conclusions

Erdheim Chester disease (ECD) is a rare, non-Langerhans cell histiocytosis of unknown etiology that occurs in multiple organs. The clinical characteristics of ECD are unknown, making it difficult to diagnose. This report described an unusual presentation of ECD involving the skeletal system and multiple extraskeletal organs. Owing to its non-specific nature, ECD was notably difficult to diagnose. Therefore, if a patient has musculoskeletal symptoms such as knee pain and other multiorgan presentations without malignancy, clinicians should suspect ECD. 

## Figures and Tables

**Figure 1 medicina-59-01288-f001:**
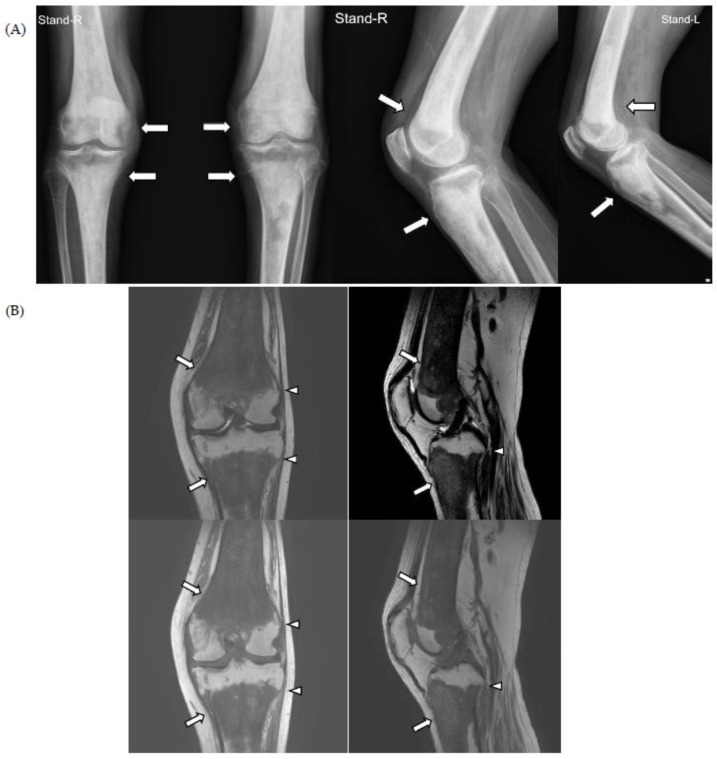
On simple radiographs of both knee (**A**) and magnetic resonance imaging (MRI) of the left knee (**B**), there is characteristic symmetrical, bilateral cortical sclerosis affecting metaphysis and diaphysis of both the distal femur and proximal tibia (white arrows). Due to the disease progression, there is patch medullary sclerosis, with resultant loss of cortico-medullary differentiation. There is a spiculated margin to the epiphysis (white arrowheads), which shows that the epiphyseal surface is relatively preserved; ECD rarely affects epiphysis and axial bones.

**Figure 2 medicina-59-01288-f002:**
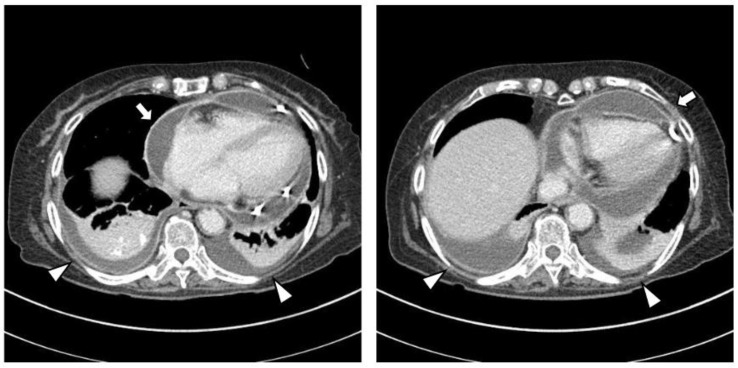
Upon chest computed tomography, there is a large amount of pericardial effusion with pericardial enhancement, which shows a thickened pericardium (white arrows). In addition, there is a moderate amount of pleural effusion and pulmonary edema with pleural enhancement (white arrowheads). There is no definite evidence of a malignant mass or nodule in either side of the lung.

**Figure 3 medicina-59-01288-f003:**
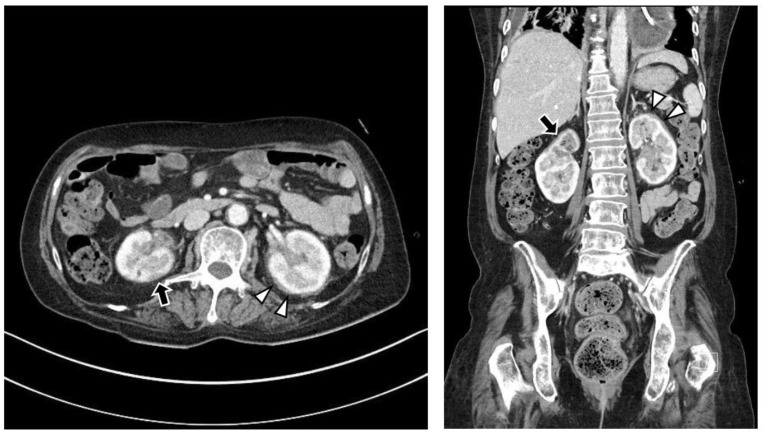
Upon abdominal computed tomography, there is fluid collection in both perirenal spaces (white arrowheads) and cortical scarring in the upper pole of the right kidney (black arrows), which is sequelae of recurrent urinary tract infection of the right kidney. There are findings of diffuse renal involvement, which are non-specific findings of medical renal disease, but there is no evidence of distant metastasis.

**Figure 4 medicina-59-01288-f004:**
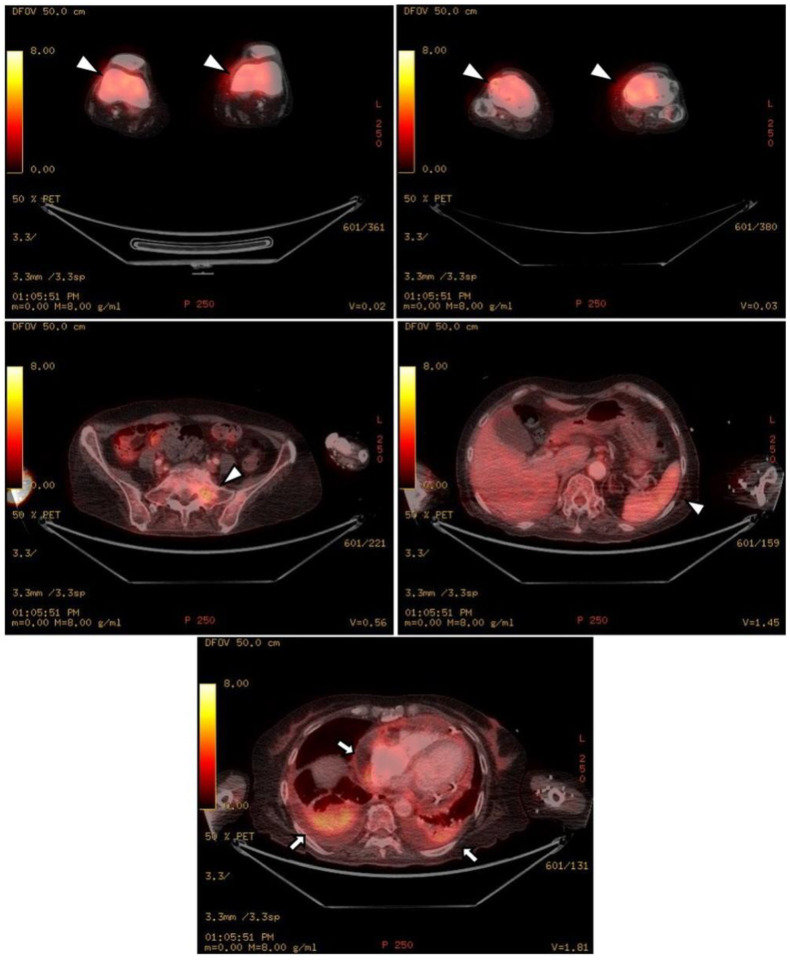
Upon PET-CT, there is diffuse FDG uptake in both distal femurs, proximal tibia, sacrum, and spleen, which shows increased metabolic activity (white arrowheads). As shown in the chest and abdominal CT, there is pericardial effusion and pleural effusion shown by FDG uptake in the pericardium and pleura (black arrows).

**Figure 5 medicina-59-01288-f005:**
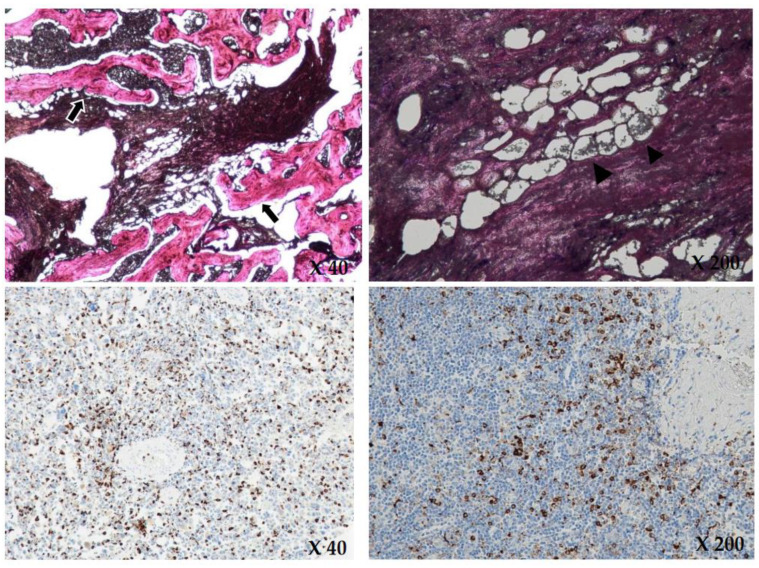
Upon H&E (hematoxylin and eosin) staining (×40, ×200), there is diffuse infiltration of foamy histiocytes (black arrowheads) and intraosseous fibrous tissue (black arrows) in the marrow space, consistent with Erdheim Chester disease. Immunohistochemical staining (×40, ×200) showed positive for CD68 and negative for CD1a and S-100. The brownish colored cytoplasms of the histiocytes in the immunohistochemical staining show positive for CD68.

**Figure 6 medicina-59-01288-f006:**
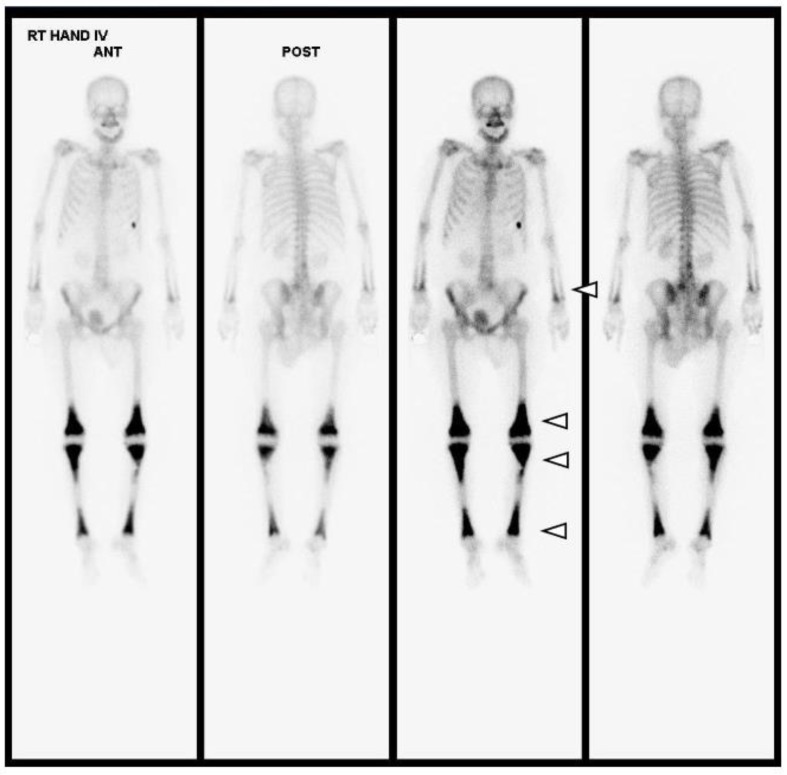
Upon whole-body bone scanning (Tc-99m HDP Bone Scintigraphy), there is intense, bilaterally symmetric uptake at the end of long bones, such as both tibias, femurs, radii and ulnas, with sparing of the epiphysis, which is characteristic of ECD (white arrowheads). Differential diagnoses of other osteosclerotic conditions include diseases such as lymphoma, chronic osteomyelitis, Paget’s disease, or metastases; however, amongst these, symmetrical metadiaphyseal uptake on nuclear scintigraphy is exclusive to ECD.

## Data Availability

The authors declare that data supporting the fundings of this study are available within the article.

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
