# Peer review of "An Unusual Case of Erdheim Chester Disease (ECD) with Knee Pain: A Case Report"

_medicina, 2023, doi:10.3390/medicina59071288_

Round 1

Reviewer 1 Report

I have reviewed the case and it looks great.It has a detailed diagnostic dilemma and diagnostic approach for very rare case . It has a great relevance for the general medicine field.

Excellent case presentation of unusual rare disease . I would suggest to add breif data on treatment options and prognosis 

Author Response

Reviewer #1

I have reviewed the case and it looks great.It has a detailed diagnostic dilemma and diagnostic approach for very rare case . It has a great relevance for the general medicine field.

  • Thank you for your comment.

Excellent case presentation of unusual rare disease . I would suggest to add breif data on treatment options and prognosis. 

  • Thank you for your comment. I have modified the article about the treatment options.
  • Line 192-205

Reviewer 2 Report

The case report is well written and well presented. The goal of the case report is clear. 

Unfortunately, the uniqueness (or novelty) of the case report is not clear to me. Comparable case reports have been documented in previous publications (References 1, 2, 3), which this paper does not acknowledge. It is essential that the paper clearly explains what makes this patient different, or what aspects of the diagnostic procedure used here are particularly unique.

Reference:

1)    Jridi M, Ben Achour T, Naceur I, et al. Erdheim-Chester disease: A multisystem disease case illustration with rare manifestations and treatment challenges. Clin Case Rep. 2023;11(2):e6996. Published 2023 Feb 24. doi:10.1002/ccr3.6996

2)    Sistermann R, Katthagen BD. Erdheim Chester disease: a rare cause of knee and leg pain. Arch Orthop Trauma Surg. 2000;120(1-2):112-113. doi:10.1007/pl00021229

3)    Rademacher S, Anagnostopoulos J, Luft FC, Kettritz R. Erdheim-Chester disease and knee pain in a dialysis patient. Clin Kidney J. 2014;7(4):402-405. doi:10.1093/ckj/sfu031 and   

Author Response

Reviewer #2

The case report is well written and well presented. The goal of the case report is clear. 

Unfortunately, the uniqueness (or novelty) of the case report is not clear to me. Comparable case reports have been documented in previous publications (References 1, 2, 3), which this paper does not acknowledge. It is essential that the paper clearly explains what makes this patient different, or what aspects of the diagnostic procedure used here are particularly unique.

Reference:

1)    Jridi M, Ben Achour T, Naceur I, et al. Erdheim-Chester disease: A multisystem disease case illustration with rare manifestations and treatment challenges. Clin Case Rep. 2023;11(2):e6996. Published 2023 Feb 24. doi:10.1002/ccr3.6996

2)    Sistermann R, Katthagen BD. Erdheim Chester disease: a rare cause of knee and leg pain. Arch Orthop Trauma Surg. 2000;120(1-2):112-113. doi:10.1007/pl00021229

3)    Rademacher S, Anagnostopoulos J, Luft FC, Kettritz R. Erdheim-Chester disease and knee pain in a dialysis patient. Clin Kidney J. 2014;7(4):402-405. doi:10.1093/ckj/sfu031 and   

  • Thank you for your comment.
  • I agree with you on that this paper may not be greatly distinguishable from other published papers. It is mainly due to that this disease itself is rare and not easily seen. Nonetheless, I would like to point out some factors that do differentiate our case to other cases. Firstly, our case had shown not only pain, but also proceeded with contracture.
  • Also, in the case of our patient, the diagnosis could have been made earlier, but did not. For knee pain and contracture, radiologic evaluation of the knee joint was performed at another hospital and the findings showed osteosclerotic lesions, which were thought to be bone metastases; therefore, the patient was referred to the hemato-oncology department of our hospital for cancer evaluation.
  • In the process of cancer evaluation, there had been a delay in consultation with the OS department in regards to knee pain. Even so, we thought it pertinent to suspect ECD at the time of consultation and actively performed bone biopsies, which may be another distinguishable factor of our case.
  • Thank you for your comment. The manuscript has been accordingly.
  • Line 206-218

Reviewer 3 Report

Owing to its non-specific nature, ECD was notably difficult to diagnose. Therefore, if a patient has knee pain and other multiorgan presentations without malignancy, clinicians should suspect ECD. Several comments given below.

1.      Nothing truly unique in its current state. Because of the lack of a novel, the current submission looks to be a replication or modified work. The authors must describe their novel in detail. This work should be rejected owing to a major issue.

2.      It is necessary to summarize previous literature' contributions, novelties, and limitations in the introduction section to highlight the work gaps that the current study aims to fill.

3.      Line 42, the statement of “This paper hopes…” seems no sound science. Please use more academic statement.

4.      Line 88, please arrange the Figure 1 into section like 1a, 1b, etc. To give more appropriate presentation.

5.      Line 95, the authors encouraged to explain the basic concept and application of computed tomography in medical purposes. Please provide the information along with relevant reference as follows: https://doi.org/10.3390/ma16093298, https://doi.org/10.1016/j.heliyon.2022.e12050, and https://doi.org/10.3390/biomedicines11020427

6.      Line 96, the differences of CT and PET-CT would giving the information for better understanding.

7.      Line 114, the authors intent to present left and right and the information differences needs to stated.

-

Author Response

Owing to its non-specific nature, ECD was notably difficult to diagnose. Therefore, if a patient has knee pain and other multiorgan presentations without malignancy, clinicians should suspect ECD. Several comments given below.

  1. Nothing truly unique in its current state. Because of the lack of a novel, the current submission looks to be a replication or modified work. The authors must describe their novel in detail. This work should be rejected owing to a major issue.
  • Thank you for your comment. The manuscript has been accordingly.
  • Line 206-218
  1. It is necessary to summarize previous literature' contributions, novelties, and limitations in the introduction section to highlight the work gaps that the current study aims to fill.
  • Thank you for your comment. The manuscript has been accordingly.
  • Line 41-43
  1. Line 42, the statement of “This paper hopes…” seems no sound science. Please use more academic statement.
  • Thank you for your comment. The manuscript has been accordingly.
  • Line 41-43
  1. Line 88, please arrange the Figure 1 into section like 1a, 1b, etc. To give more appropriate presentation.
  • Thank you for your comment. The manuscript has been accordingly.
  •  
  1. Line 95, the authors encouraged to explain the basic concept and application of computed tomography in medical purposes. Please provide the information along with relevant reference as follows: https://doi.org/10.3390/ma16093298, https://doi.org/10.1016/j.heliyon.2022.e12050, and https://doi.org/10.3390/biomedicines11020427
  • Thank you for your comment.
  • The term “computed tomography,” or CT, refers to a computerized x-ray imaging procedure in which a narrow beam of x-raysis aimed at a patient and quickly rotated around the body, producing signals that are processed by the machine’s computer to generate cross-sectional images, or “slices.” These slices are called tomographic images and can give a clinician more detailed information than conventional x-rays.
  • However, It is considered unnecessary to write these explanations in this case report.
  1. Line 96, the differences of CT and PET-CT would giving the information for better understanding.
  • Thank you for your comment.
  • The biggest difference between CT scans and PET scans is that PET scans measure metabolic activity, while CT scans create detailed images of the body’s organs and tissues.
  • PET scans are also more sensitive than CT scans, which means they can be used to detect cancer at an early stage.
  • However, It is considered unnecessary to write these explanations in this case report, too.
  1. Line 114, the authors intent to present left and right and the information differences needs to stated.
  • Thank you for your comment. The manuscript has been accordingly

 Line 117

Round 2

Reviewer 2 Report

Based on my recommendations, the paper has been successfully revised and now fulfills all the necessary criteria for publication in Medicina.

Author Response

Reviewer #2

Based on my recommendations, the paper has been successfully revised and now fulfills all the necessary criteria for publication in Medicina.

  • Thank you for your thoughtful comment.

Reviewer 3 Report

Reviewers greatly appreciate the efforts that have been made by the author to improve the quality of their articles after peer review. I reread the author's manuscript and further reviewed the changes made along with the responses from previous reviewers' comments. Unfortunately, the authors failed to make some of the substantial improvements they should have made making this article not of decent quality with biased, not cutting-edge updates on the research topic outlined. In addition, the author also failed to address the previous reviewer's comments, especially on comments number 1 (Nothing new), (Poor explanation), and 5 (not incorporated the literature). Thank you very much for the opportunity to read the author's current work.

-

Author Response

Reviewers greatly appreciate the efforts that have been made by the author to improve the quality of their articles after peer review. I reread the author's manuscript and further reviewed the changes made along with the responses from previous reviewers' comments. Unfortunately, the authors failed to make some of the substantial improvements they should have made making this article not of decent quality with biased, not cutting-edge updates on the research topic outlined. In addition, the author also failed to address the previous reviewer's comments, especially on comments number 1 (Nothing new), (Poor explanation), and 5 (not incorporated the literature). Thank you very much for the opportunity to read the author's current workThank you for your comment. The manuscript has been accordingly.

  • Thank you for your comment.
  • As you mentioned, I have referred back to updated papers on the latest research topics and made revisions to the article.
  • As for the point you mentioned as number 1, I will provide an explanation
  • I agree with you on that this paper may not be greatly distinguishable from other published papers. It is mainly due to that this disease itself is rare and not easily seen. Nonetheless, I would like to point out some factors that do differentiate our case to other cases. Firstly, our case had shown not only pain, but also proceeded with contracture.
  • Also, in the case of our patient, the diagnosis could have been made earlier, but did not. For knee pain and contracture, radiologic evaluation of the knee joint was performed at another hospital and the findings showed osteosclerotic lesions, which were thought to be bone metastases; therefore, the patient was referred to the hemato-oncology department of our hospital for cancer evaluation.
  • In the process of cancer evaluation, there had been a delay in consultation with the OS department in regards to knee pain. Even so, we thought it pertinent to suspect ECD at the time of consultation and actively performed bone biopsies, which may be another distinguishable factor of our case.
  • Thank you for your comment. The manuscript has been accordingly.
  • Line 207-219
  •  
  • Furthermore, in response to criticism from number 5, I have offered further clarifications regarding the purpose of CT scans.
  • Line 98
  • I also appreciate the papers you have pointed out for me to refer to. Thank you.